# Tip Lesion Most Frequent FSGS Variant Related to COVID-19 Vaccine: Two Case Reports and Literature Review

**DOI:** 10.3390/vaccines12010062

**Published:** 2024-01-08

**Authors:** Emmy Marjorie Carvalho de Araújo, Marcos Adriano Garcia Campos, Andressa Monteiro Sodré, Maria Izabel de Holanda, Rodrigo Hagemann, Antonio Augusto Lima Teixeira Júnior, Natalino Salgado Filho, Precil Diego Miranda de Menezes Neves, Gyl Eanes Barros Silva

**Affiliations:** 1Faculty of Medicine, Federal University of Maranhão, Gonçalves Dias Square, São Luís 65020-240, Brazil; marjorie.emmy@discente.ufma.br (E.M.C.d.A.); andressa.ms@discente.ufma.br (A.M.S.); natalinosalgadofilho@uol.com.br (N.S.F.); 2Clinical Hospital of Botucatu Medical School, São Paulo State University, Professor Mário Rubens Guimarães Montenegro Avenue, Botucatu 18618-687, Brazil; mag148@duke.edu; 3Bonsucesso Federal Hospital, Londres Avenue, Rio de Janeiro 21041-020, Brazil; izabeldeholanda@gmail.com; 4Clinical Hospital Complex, Federal University of Paraná, General Carneiro Street, Curitiba 80060-900, Brazil; rodrigo.hagemann@ufpr.br; 5Departament of Genetics, Ribeirão Preto Medical School, University of São Paulo, Bandeirantes Avenue, Ribeirão Preto 14040-900, Brazil; aaltjr@usp.br; 6Clinical Hospital, University of São Paulo, Dr. Ovídio Pires de Campos Street, São Paulo 05403-010, Brazil; pmenezes@haoc.com.br; 7Department of Pathology and Legal Medicine, Ribeirão Preto Medical School, University of São Paulo, Bandeirantes Avenue, Ribeirão Preto 14040-900, Brazil

**Keywords:** vaccine, COVID-19, SARS-CoV-2, focal segmental glomerulosclerosis, kidney biopsy

## Abstract

Large-scale COVID-19 vaccination has been one of the most effective strategies to control the spread of the SARS-CoV-2 virus. However, several cases of glomerular injury related to the COVID-19 vaccine have been described in the literature. We report two cases of a tip lesion variant of focal segmental glomerulosclerosis (FSGS), which presented with significant proteinuria and improved after immunosuppression. In our literature review, the tip lesion variant of FSGS is currently the most frequent variant associated with vaccination against COVID-19. Prognosis is favorable and without significant alterations in the tubulointerstitial or vascular compartments. Adverse effects of vaccines need to be recognized early and will help us to understand the immune and pathological mechanisms of kidney damage.

## 1. Introduction

Coronavirus disease 2019 (COVID-19), caused by the SARS-CoV-2 virus, is highly contagious and can be fatal. One of the most effective public health strategies for containing this virus and minimizing the risk of contagion is large-scale vaccination [1]. The genetic sequencing of the virus allows researchers to study the development of live attenuated virus vaccines, inactivated virus vaccines, non-replicating viral vector vaccines, and mRNA vaccines using different technologies [2,3].

However, after the massive implementation of these vaccines, there have been several reports of new or reactivated glomerular diseases, especially in patients with exacerbated immune responses to immunization, such as those with immunoglobulin A nephropathy (IgAN), minimal change disease (MCD), and focal segmental glomerulosclerosis (FSGS), including FSGS tip lesion variant [4,5,6].

This study describes the development of FSGS tip lesion variant in patients who received a vaccine against COVID-19. We performed a literature review with articles related to post-COVID-19 FSGS by searching on PubMed and Google Scholar since 2020 for the following terms: “COVID-19 Vaccines” and “Glomerulonephritis” or “Nephrotic Syndrome” or “Glomerulosclerosis, Focal Segmental”.

## 2. Case Reports

### 2.1. Case 1

A 47-year-old, Caucasian man with hypertension and depression complained of progressive facial edema three weeks after receiving the first dose of the Oxford–AstraZeneca (AZ) vaccine. During hospitalization, the disease progressed to anasarca, which was associated with reduced diuresis, hypertension, hematuria, and progressive worsening of his renal function.

Upon admission, the patient’s creatinine was 2.6 mg/dL, urine protein was 12,673 mg/24 h, albumin was 1.7 mg/dL, C3 was 171 mg/dL, and C4 was 50 mg/dL. Antinuclear antibody (ANA) and antineutrophil cytoplasmic antibodies (ANCAs) were not measured. The patient’s serology was negative. Methylprednisolone pulse therapy was initiated. After three weeks of hospitalization, the patient was transferred to a tertiary service where he received oral prednisone (1 mg/kg/day), and a renal biopsy was performed after five days of pulse therapy. On that occasion, the patient had a negative reverse transcription polymerase chain reaction (RT-PCR) for COVID-19. But a new test was carried out after a week, and the patient presented a positive RT-PCR for COVID-19 during hospitalization.

The renal biopsy revealed six glomeruli that were normal except for the presence of a glomerulus with segmental sclerosis and synechia in the tubular pole. Only part of each glomerulus was damaged. The arterioles were normal. The tubulointerstitium was preserved. Immunofluorescence revealed moderately segmental deposition for C3 and immunoglobulin M in the vascular poles of some glomeruli. The patient was diagnosed with FSGS tip lesion variant. Genetic testing of high-risk alleles of the APOL1 gene revealed G0/G0.

The patient was treated with prednisone (1 mg/kg/day), furosemide, spironolactone, rivaroxaban, omeprazole, calcium carbonate, and cholecalciferol. After one year, the patient had a creatinine of 0.9 mg/dL, an albumin of 4.1 mg/dL, and a protein/creatinine ratio of 0.06. The prednisone was discontinued, and the patient was maintained on cyclosporine due to corticosteroid dependence.

### 2.2. Case 2

A 17-year-old woman without comorbidities complained of moderate-intensity abdominal pain five days after receiving the Pfizer–BioNTech vaccine for COVID-19. The patient was initially diagnosed with a urinary tract infection. Levofloxacin was administered for five days, though the patient presented with lower limb edema with a progressive increase in eyelid edema after 10 days. No renal abnormalities were observed on ultrasound.

The patient had a creatinine of 2.2 mg/dL, urine protein of 9125 mg/24 h, hemoglobin of 12.7 g/dL, and hematocrit of 36.3%. A urinalysis revealed the absence of hematuria, leukocyturia, and casts. Serological test results for HIV, HCV, and HBV were negative. ANA and ANCA tests were not performed for this patient.

A renal biopsy revealed 18 glomeruli and five arteries. Twelve of the glomeruli were normal and six had segmental sclerosis, synechia, and hyaline deposits, all in the tubular pole. The damage was segmental because it affected only a part of the glomerulus. All of the arteries were within the normal limits. Interstitial tubules with mild degenerative changes were observed. Twelve glomeruli underwent immunofluorescence analysis, with segmental deposition for C3 and immunoglobulin M deposits in the vascular poles of some glomeruli. Transmission electron microscopy of the paraffin-embedded tissue shows diffuse effacement of foot processes as well as microvillous transformation but no immune-type deposits. The patient was diagnosed with FSGS tip lesion variant associated with mild acute tubular necrosis (Figure 1). The protocol used for histopathological analysis is available in the Appendix A.

Therefore, pulse therapy with prednisone (1 mg/kg) was initiated. The patient presented with a complete response in less than four weeks. The creatinine improved to 0.8 mg/dL and no proteins were detected in the urine. The prednisone treatment continued for approximately eight months, including the weaning period.

## 3. Discussion

### 3.1. Mechanisms of Post-Vaccine Glomerular Injury

The main coronavirus vaccines used during the pandemic were non-replicating viral vector and mRNA vaccines [3]. Non-replicating viral vector vaccines contain modified adenoviruses and are replication-defective. For example, the AZ vaccine uses a modified chimpanzee adenovirus. mRNA vaccines use lipid nanoparticles to transport genetically modified mRNA to the cell cytoplasm without entering the nucleus, which is a safe method with regard to the risk of adverse effects. The most commonly used mRNA vaccines are the BNT162b2 COVID-19 (Pfizer–BioNTech, Mainz, Germany) and mRNA-1273 (Moderna, Cambridge, MA, USA) vaccines [7].

However, the development of post-vaccination adverse effects is not new. Since the implementation of mass vaccination, several short-term adverse effects have been reported at the application site (pain, inflammation, and urticarial eruptions) or systemically (myocarditis, cognitive disorders, and kidney damage) [8,9,10]. Acute kidney injury (AKI) and glomerulopathies have been associated with COVID-19 vaccines in several reports [11]. A series of thirteen patients in whom glomerulonephritis was triggered after mRNA vaccination included eight patients who developed de novo glomerulonephritis and five who had relapses [12].

The immune response to the COVID-19 vaccine may be a trigger of nephropathies caused by T cell- and B cell-mediated podocyte damage, the humoral and cellular immune response mediated by the mRNA vaccine, and the genetic predisposition of the individual. Usually, renal involvement occurs in less than 20 days after the first injection, with the temporal relationship being smaller in subsequent injection doses [13]. Types of glomerulonephritis included IgAN, membranous nephropathy (MN), and primary podocytopathies, such as MCD and FSGS (mainly the tip lesion variant), and were related to ANCA [12,14,15].

Post-SARS-CoV-2 vaccination IgAN was reported in 48 patients, including 31 new cases and 17 relapses. Thirty of the patients had received the Pfizer vaccine, fifteen had received the Moderna vaccine, and three had received the AZ vaccine [5]. Currently, the most accepted theory related to the pathogenesis of IgAN is the multi-hit theory, in which triggering events, such as infectious or inflammatory conditions, precede the development of the disease [6]. There are several reports of adult and pediatric patients who developed IgA after vaccination, which may correspond to previously unrevealed glomerulonephritis or the vaccination triggering disease in patients with deposited IgA. However, the mechanisms associated with vaccination are not fully understood [16].

In addition to IgAN, MN has been associated with the COVID-19 vaccine, including new and recurrent cases [17,18]. In a study of 210 patients with a previous diagnosis of MN who were immunized with the Pfizer vaccine, 6 relapsed [19]. Immune dysregulation, which leads to non-identification and intolerance to podocyte antigens, is described as a possible cause of the onset or reappearance of nephropathy, although the mechanism has not yet been fully elucidated [20].

### 3.2. Nephrotic Syndrome after COVID-19 Vaccine

The action of vaccines as triggers for the development or reactivation of nephrotic syndrome has been reported for vaccines against meningococcus C and B viruses, influenza, and diphtheria pertussis [21]. Some forms of podocytopathy have also been associated with COVID-19 vaccines. Kronbichler et al. suggested that there is a temporal and pathogenic association between vaccination and podocytopathies, especially MCD, after the first dose and between other podocytopathies after the second dose [22]. In a recent systematic review about nephrotic syndrome following COVID-19 immunization, pathogenesis was mainly attributed to the activation of angiotensin-converting enzyme-2 receptors, which leads to podocyte effacement [23].

The first description of the association between vaccination and the development of MCD was related to the influenza virus and highlighted the acute onset of the disease after vaccination [24,25]. More recently, prospective studies have reported a low rate of relapse in children who receive influenza vaccinations that is similar to the rate of relapse among unvaccinated children. Although external factors or coincidental events cannot be excluded, the short temporal relationship and the onset or reappearance of renal syndromes suggest a possible pathogenic correlation [26]. Several reports have reported the appearance of MCD soon after the administration of the Pfizer–BioNTech vaccine, including that in a previously healthy patient who developed MCD with nephrotic syndrome and AKI after the first dose of the vaccine [27,28]. The short temporal relationship and the exclusion of other previous factors suggests an association between the development of nephropathies and vaccination [27,29].

In addition to the vaccine, COVID-19 can affect the kidneys. Renal impairment is a potential symptom of SARS-CoV-2 infection, leading to increased morbidity and mortality. However, glomerular immunization reactions do not follow the same pattern as those related to infection [30]. Recently, a COVID-19-related podocyte lesion morphologically compatible with collapsing glomerulopathy (CG) was termed COVID-19-associated nephropathy (COVAN), though its etiology remains unclear [29,30,31]. A systematic review of 59 studies indicated that CG is the most prevalent glomerular histopathological finding related to SARS-CoV-2 [32,33], demonstrating that although viral infection favors the appearance of collapsing podocytopathy, patterns of changes after vaccination are different, as the development of MCD and other forms of FSGS are much more frequent than CG.

There are few reports of CG after vaccination, including post-vaccine CG, after Pfizer and AZ, in two patients who developed pulmonary congestion, dyspnea, massive edema, and chronic kidney disease (CKD) [34]. Another patient was diagnosed with CG after receiving the first dose of the Pfizer vaccine. The patient was healthy before receiving the vaccine and developed disease exactly seven days after vaccination, coinciding with the peak production of follicular T cells stimulated by the mRNA vaccine [35]. In kidney transplantation, a case of GC after mRNA (Moderna) COVID-19 vaccination was also found [36].

### 3.3. FSGS Tip Lesion Variant after COVID-19 Vaccine

Non-collapsing forms of FSGS are frequently associated with vaccination against SARS-CoV-2 [12,21]. The disease is morphologically classified into five variants according to the Columbia system: cellular, perihilar, collapsing, not otherwise specified (NOS), and tip lesion. The tip lesion variant presents as a sclerotic lesion at the tip of the glomerular tuft at the origin of the proximal tubule, with or without an excess of mesangial matrix and adhesion between the tuft and Bowman’s capsule (synechiae). It is characterized by higher levels of proteinuria and lower serum albumin levels than perihilar and NOS variants. Patients with the tip lesion variant typically have a good prognosis [37,38,39]. The frequency of the tip lesion variant is up to 17% and it may present with lower tubulointerstitial involvement [37,40,41]. However, among 14 patients with post-vaccine FSGS (Table 1), 7 were classified as having the tip lesion variant. The frequency may be higher as some described cases of FSGS are not classified.

FSGS has been characterized as one of the three main glomerulopathies associated with vaccination against COVID-19, and the tip lesion variant is the most common [12,51]. The patients presented in this report developed this variant after vaccination with the Pfizer or AZ vaccines. FSGS tip lesions are more commonly due to the Pfizer vaccine (Table 2). This may be due to the greater accessibility of the Pfizer vaccine and the greater efficiency in the synthesis of the COVID-19 spike protein [52]. However, factors such as sex, age, comorbidities, immunosuppressive status, and possible infection may also influence the appearance of these lesions [53].

The percentage of patients with adverse symptoms after vaccination with the Pfizer vaccine is 79% after the first dose and 84% after the second dose [54]. The development of renal complications, more specifically tip lesions, occurs after the second dose, with a greater number of new cases than relapses. This may be due to the increase in the response of the innate immune system after the second dose compared to the primary immunization, as the second dose of mRNA can induce a storm of cytokines including interferon-γ and CXCL10 [55,56].

Another important COVID-19 post-vaccine diagnosis is the development of autoimmune diseases like systematic lupus erythematous (SLE). The mechanism of the post-vaccine SLE-like syndrome may be due to an inactive viral component, vaccine adjuvants, or an attenuated microbe that generates molecular mimicry or bystander activation in genetically susceptible patients. Clinical manifestations, including renal involvement, and ANA can help with the diagnosis [57].

Most patients who develop the tip lesions variant are adult, Caucasian men between 20 and 45 years of age [58]. The mean age of patients with reported post-vaccination FSGS tip lesion variant is 36 years (range: 17–67 years), and the number of males and females is equal. Data regarding patient ethnicity have not been reported in most cases of post-vaccine FSGS tip lesion variant. Furthermore, no comorbidities were significantly associated with the development of post-immunization FSGS. One of the two patients presented in this study had a comorbidity (arterial hypertension).

According to the Kidney Disease Improving Global Outcomes guidelines for the management of patients with glomerulopathies, the treatment of FSGS is immunosuppressive therapy, including a combination of corticosteroids, calcineurin inhibitors, and cytotoxic agents [58,59]. In this study, the first patient was treated with prednisone and cyclosporine and experienced partial remission and slow evolution with the recurrence of nephrotic proteinuria. The second patient was treated with corticosteroids (prednisone) and experienced complete symptom remission within a few weeks. In previous studies, patients with post-vaccination tip lesion FSGS were also treated with immunosuppressants, such as cyclophosphamide, corticosteroids, and tacrolimus (Table 2). In addition to corticosteroid therapy, Janus kinase inhibitors appear to be a promising treatment for relapsing tip lesion variant FSGS [60]. One patient with FSGS non-tip variant was explicitly maintained on hemodialysis, confirming the low rate of progression to renal replacement therapy (RRT) among patients with the tip variant [59,60].

In general, the tip lesion variant presents with less severe hyalinosis and interstitial fibrosis or mild to moderate tubular atrophy when compared to other variants of FSGS, such as CG [41,61]. In addition, it is associated with significant renal survival. A retrospective study of 116 patients with primary FSGS reported that six of nine patients with the tip lesion variant had alterations in the tubulointerstitial compartment and none had fibrosis [41]. In our case report, the preservation of the tubulointerstitial compartment was observed in both patients. No vascular alterations or fibrosis were observed. Therefore, the patients in this study had a normal pattern of disease.

## 4. Conclusions

The tip lesion variant of FSGS may be the most frequently reported variant described in the literature associated with vaccination against COVID-19, mainly mRNA vaccines and rarely non-replicating viral vector vaccines. In general, the prognosis is favorable, with rare reports of evolution to RRT and without significant alterations in the tubulointerstitial or vascular compartments. The epidemiological profile is similar to that of patients with tip lesions of primary origin. Physicians must be aware of the possible nephrological effects of vaccines to recognize and understand the immune and pathological mechanisms and potential effects on the body at an early stage.

## Figures and Tables

**Figure 1 vaccines-12-00062-f001:**
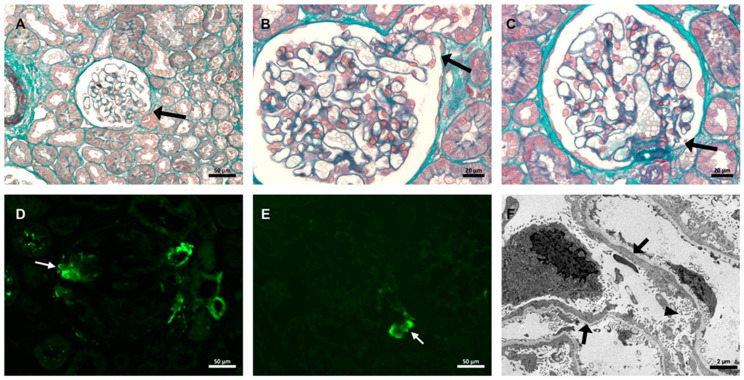
Histopathological findings in renal biopsy. (**A**) Glomerulus with segmental sclerosis at the tubular pole (black arrow) surrounded by normal tubulointerstitial compartment. (**B**) Segmental sclerosis with adherence and herniated into the tubular lumen (black arrow). (**C**) Normal glomerulus in the same patient (black arrow indicating vascular pole). (**A**–**C**) Masson’s trichrome stain. (**D**,**E**) C3 and IgM deposits in tubular pole observed via immunofluorescence (white arrow). (**F**) Electron microscopy showing diffuse podocytes foot process effacement (black arrow) and microvillous transformation (black head arrow).

**Table 1 vaccines-12-00062-t001:** Segmental and focal glomerulosclerosis after vaccination against COVID-19.

Author	N	Variant
Hummel et al. (2022) [42]	2	Tip lesion
Fenoglio et al. (2022) [13]	1	Tip lesion
Klomjit et al. (2021) [12]	1	Tip lesion
Timmermans et al. (2022) [4]	1	Tip lesion
Huang et al. (2022) [43]	1	Not specified
Sirpal et al. (2022) [44]	1	Tip lesion
Vijoy Kumar Jha et al. (2022) [45]	1	Tip lesion
Cho A Lim et al. (2022) [35]	1	Cellular
Horinouchi et al. 2023 [46]	1	Collapsing
Marega et al. (2022) [47]	1	Collapsing
Khan et al. (2022) [48]	1	Collapsing
Rajarathinam et al. (2022) [49]	2	Not specified
Roy et al. (2022) [50]	1	Not specified

**Table 2 vaccines-12-00062-t002:** Characteristics of patients with segmental and focal glomerulosclerosis tip lesion variant after COVID-19 vaccination.

Author	Sex	Age (Years)	Vaccine	Dose	Days until Symptoms	Proteinuria (Initial/Final)	Creatinine (Initial/Final)	Albumin (Initial/Final)	Type of Case	Treatment	RRT
Hummel et al. (2022) [42]	Female	46	Pfizer	2	20	0.1 g/g/9 g/g	2.1 mg/dL/-	3.9 g/dL/2.2 g/dL	Relapse	CTX	-
Hummel et al. (2022) [42]	Female	67	Pfizer	1	18	0.3 g/g/9.2 g/g	3.7 mg/dL/-	4.0 g/dL/1.9 g/dL	Relapse	CsA + MMF	-
Fenoglio et al. (2022) [13]	Female	24	Pfizer	2	88	-/-	-/-	-/-	De novo	GC	No
Klomjit et al. (2021) [12]	Female	29	Pfizer	2	21	-/10 g/dL	0.6 mg/dL/0.7 mg/dL	2.2 g/dL/3.2 g/dL	Relapse	GC + FK	No
Timmermans et al. (2022) [4]	Male	47	Pfizer	2	14	-/4.5 g/d	0.8 mg/dL/-	2.0 g/dL/-	De novo		-
Sirpal et al. (2022) [44]	Male	26	Moderna	2	21	-/3.2 g (24 h)		5 g/dL/2.7 g/dL	De novo	FK	-
Vijoy Kumar Jha et al. (2022) [45]	Male	21	AZ	1	12	23.87 g (24 h)/20.7 g (24 h)	1.06 mg/dL/1.02 mg/dL	0.9 g/dL/-	De novo	GC + FK	-
Patient 1 *	Male	47	AZ	1	30	12.6 g (24 h)/normal	2.6 mg/dL/0.91 mg/dL	1.7 g/dL/4.1 g/dL	De novo	GC + CsA	No
Patient 2 *	Female	17	Pfizer	3	5	9.125 g/dL/normal	2.2 mg/dL/0.8 mg/dL	-/-			

AZ: Oxford AstraZeneca vaccine; CTX: Cyclophosphamide; CsA: Cyclosporine; MMF: Mycophenolate mofetil; GC: Glucocorticoid; FK: Tacrolimus; RRT: renal replacement therapy; * Patient described in the current study.

## Data Availability

Data are contained within the article and Appendix A.

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
