# Peer review of "Tip Lesion Most Frequent FSGS Variant Related to COVID-19 Vaccine: Two Case Reports and Literature Review"

_vaccines, 2024, doi:10.3390/vaccines12010062_

Round 1
Reviewer 1 Report (Previous Reviewer 4)
Comments and Suggestions for Authors
All comments were addressed
Author Response
Please see the attachment.

Reviewer 2 Report (New Reviewer)
Comments and Suggestions for Authors
In the manuscript entitled “Tip lesion most frequently FSGS variant related to COVID-19
vaccine: two case report and literature review” the authors present two cases of FSGS tip lesion variant following COVID-19 vaccination. They also review the literature about the development of glomerular diseases post COVID-19 vaccination, focusing on FSGS tip lesion variant. In my opinion, it is a well-writen, comprehensive work that the readers will find educative. I have few suggestions:
1. The title could be better rephrased, for example “Tip lesion is the most frequent FSGS variant related to COVID-19 vaccine: two case reports and literature review”
2. Spelling mistakes could be corrected and language could be improved in few occasions, eg segmental and focal segmental (line 44), positive C3 segmental immunoglobulin M (line 66, line 89), related ANCA (line 129)
3. It would be interesting if the authors commented on the fact that patient 1 was found COVID-19 positive on diagnosis of nephrotic symptom. Could renal disease be associated with COVID-19 disease and not vaccination? Was he still positive when steroid pulses were administered, even before renal biopsy was performed?
4. In immunofluorescence both patients were positive for C3 and IgM deposition. What about the other patients with FSGS tip lesion mentioned in the review? Are there data on immunofluorescence findings?
5. Is renal impairment indeed a main symptom of COVID-19 disease, as stated in line 164?
Comments on the Quality of English LanguageNeeds moderate English language editing
Author Response
Please see the attachment.

Reviewer 3 Report (New Reviewer)
Comments and Suggestions for Authors
Major comments:
Although the author's report seems perfect in the case description but needs to add the following answers to be more MS-impacted
1. Your literature review depends on which mechanism of search. Which databases do you search (PubMed/Medline, Embase, Scopus, web of Science, Cochrane or Google Scholar databases? Which keywords did you use? Which time interval do you use? All of these should be added under the method section.
2. The perfect results in Figure 1, who you did, harvest, you should mention the methodology with reagent, and references used.
3. You should have mentioned in detail, if both cases have both focal and/or segmental glomerulosclerosis, because there are standard differences between both of them, please refer to a. https://www.kidneypathology.com/English_version/Histologic_patterns.html#:~:text=Glomerular%20changes%20can%20be%20focal,the%20extension%20of%20the%20affectation, b. https://www.pennmedicine.org/for-patients-and-visitors/patient-information/conditions-treated-a-to-z/focal-segmental-glomerulosclerosis#:~:text=%22Focal%22%20means%20that%20some%20of,an%20individual%20glomerulus%20is%20damaged.
4. In Table 1, you mentioned 13 previous reports; in Table 2, you only mentioned 8 reports including your current cases. Why? Within Table 2 please add a column for which type of tip lesion variant of focal or/and segmental glomerulosclerosis.
5. All your mechanism behind these pathogenies discussion is dedicated to the mRNA vaccine, but what about AZV, which your report considered the first worldwide for your knowledge?
6. Try to widen your review of the literature scope (within the pandemic) to include reports like this (PMID: 37274308, 34924204, 37505405, 10224756, 34741283).
7. Minor comment in the conclusion or other MS parts when you mentioned the FSGS post vaccination please identify the vaccine type mRNA, AZV…
Author Response
Please see the attachment.

Reviewer 4 Report (New Reviewer)
Comments and Suggestions for Authors
The authors showed two cases with FSGS tip lesion variant after the vaccination against COVID-19.
Please explain the features of new or reactivated glomerular diseases after vaccinations against COVID-19. Please describe which type of glomerular diseases is dominant.
In Introduction section, the explanation of FSGS tip lesion variant should be added.
Discussion section is too long. Please shorten the length of Discussion section.
In line 211, “LES” would be “SLE”.
Comments on the Quality of English LanguageMinor editing of English language required.
Round 2
Reviewer 3 Report (New Reviewer)
Comments and Suggestions for Authors
Thank you. You need to add the definition for both (comments 3) according to the two links sent for you. Not all readers are pathologists, nephrologists, and other physicians.
Author Response
Thank you. You need to add the definition for both (comments 3) according to the two links sent for you. Not all readers are pathologists, nephrologists, and other physicians.
Thank you for the comment. We include this information on each case (highlighted in yellow).
This manuscript is a resubmission of an earlier submission. The following is a list of the peer review reports and author responses from that submission.
Round 1
Reviewer 1 Report
Comments and Suggestions for Authors
Dear authors
thanks for this extensive review about the COVI-19 and vaccines against it possible association with glomerular disease. But I noticed that the cases that you presented didn’t have clear signals for the presumed association especially the clinical information and the temporal association with either vaccination or COVID-19 disease itself isn’t clear from your presentation.
Author Response
Dear Reviewer, Thank you for your consideration. As shown in our table 2, the time between immunization and the presentation of symptoms can vary and is not completely defined in the literature. Some case series showing a close temporal relationship (17.5 days after a first injection, 12 days after a second injection and 4.5 days after a third injection) was observed between vaccination against COVID-19 and nephrotic syndrome. In most studies in the literature, the diagnosis is made after excluding other secondary causes. Indeed, The immune response to the COVID-19 vaccine may be a trigger of nephropathies caused by T cell- and B cell-mediated podocyte damage, the humoral and cellular immune response mediated by the mRNA vaccine and genetic predisposition of the individual. We have added this information in the discussion text for better understanding (highlighted in red).
The authors.
Reviewer 2 Report
Comments and Suggestions for Authors
The review is well-written and complete. The two new cases are well studied but their study lacks the electron microscopy evaluation that is necessary in kidney pathology to confirm the diagnosis and to have precious data. In fact, electron microscopy is considered a mainstay in the analysis of renal biopsies where it is typically employed in a correlative fashion along with light and immunofluorescence microscopy. For example, the NEPTUNE study, which is an ongoing multicentre prospective observational cohort study of children and adults with proteinuria and where only patients with minimal change disease (MCD), FSGS, or MN are eligible, renal biopsy tissue stained glass slides are scanned and uploaded in the NEPTUNE Digital Pathology Repository as whole slide images (WSI) but together with the digital electron microscopy images. So the authors should include an electron microscopy study which is optimal in fresh tissue fixed in glutaraldehyde but is possible also from paraffin-embedded tissue nonetheless with suboptimal results.
Author Response
Dear Reviewer,
Thank you for your consideration and for the suggestion to add an electron microscopy study of the kidney biopsy. We performed electron microscopy fixed in paraffin-embedded tissue of one of the reported cases and added it in Figure 1 of the article.
The authors.
Reviewer 3 Report
Comments and Suggestions for Authors
Carvalho de Araujo EM and coauthors report two cases of focal segmental glomerulosclerosis, tip lesion variant following SARS-CoV2 vaccination. The patients improved after treatment with steroids and immunosuppressive drugs. The authors reviewed the literature on this topic.
The case report is not a novelty. There are several papers already published on the occurrence of glomerular disease after Covid-19 vaccine. The most important issue is to demonstrate with certainty the pathogenetic role of the Covid-19 vaccine in inducing glomerular disease. Furthermore the authors do not even show images of the kidney biopsy performed in the two patients. Also their conclusion that tip lesion is the most frequent variant associated with vaccination against COVID 19 is not supported by scientific data. I would have said it may be the most frequently reported variant described in literature.
Author Response
Dear Reviewer,
Thank you for the considerations. We add Figure 1 with histological, immunohistochemical, immunofluorescence and electron microscopy images of the renal biopsy of one of the cases described. We changed the conclusion as suggested (highlighted in red).
The authors.
Reviewer 4 Report
Comments and Suggestions for Authors
In this case series the Author reported two cases of FSGS after related to SARS CoV2 vaccine.
The title is informative and both cases are described appropriately. The review appear to be exhaustive.
MINOR COMMENT
- AS tha ANA were not assessed the Authors should briefly discuss the possibility a post-vaccine LES-like syndrome. Moreover some parameters were not assessed or monitored (i.e. C3, C4, nDNA Ab, CRP, Hb, WBC, PLT...).
Author Response
Dear Reviewer,
Thank you for the considerations. An exhaustive investigation of secondary causes of Tip lesion FSGS variant was performed in both reported cases, including investigation for post-vaccine lupus-like syndrome. We have improved the writing of the discussion to make this information clear and exploring the possibility of post-vaccine lupus-like syndrome (highlighted in red).
The authors.
Round 2
Reviewer 1 Report
Comments and Suggestions for Authors
I believe that COVID-19 is a strange virus to humanity and caused us all a lot of confusion espcially with mutiple presentations, adding to that its vaccine espcially the mRNA vaccine and its potential to cause autoimmune processes and unpredicated complications like TTP picutre and others.
The renal involvment is one of the above possible associations. I appreciate your work.
Author Response
Reviewer
I believe that COVID-19 is a strange virus to humanity and caused us all a lot of confusion espcially with mutiple presentations, adding to that its vaccine espcially the mRNA vaccine and its potential to cause autoimmune processes and unpredicated complications like TTP picutre and others.
The renal involvment is one of the above possible associations. I appreciate your work.
Answer
Dear Reviewer,
All authors are grateful for the comment. We also believe in this association, but the literature is still controversial in some aspects. We believe that our reported cases and our literature review can contribute to a better understanding. Thanks so much for the review process.
The authors.
Reviewer 2 Report
Comments and Suggestions for Authors
at row 98 effacement better than deletion
Author Response
Reviewer
At row 98 effacement better than deletion
Answer
Dear Reviewer,
All authors are grateful for the suggestion. We replace the word "deletion" with "effacement" at row 98.
The authors.